



# Brief communication: Improving ERA5-Land soil temperature in permafrost regions using an optimized multi-layer snow scheme

Bin Cao[1], Gabriele Arduini[2], and Ervin Zsoter[2,3]

[1]National Tibetan Plateau Data Center (TPDC), State Key Laboratory of Tibetan Plateau Earth System, Environment and Resources (TPESER), Institute of Tibetan Plateau Research, Chinese Academy of Sciences, Beijing, China
[2]European Centre for Medium-Range Weather Forecasts, Reading, UK
[3]Department of Geography and Environmental Science, University of Reading, Reading, UK

**Correspondence:** Bin Cao (bin.cao@itpcas.ac.cn)

**Abstract.** We previously reported a notable warm bias in ERA5-Land soil temperature in permafrost regions that was supposedly being caused by an underestimation of the snow density. In this study, we implemented and evaluated a new multi-layer snow scheme in the land surface scheme of ERA5-Land, i.e., HTESSEL, with revised snow densification parametrizations. We compared permafrost soil temperatures from the numerical experiments with observations and the original ERA5-Land with a

single layer snow scheme. The revised HTESSEL significantly improved the representation of soil temperature in permafrost regions compared to ERA5-Land. The daily warm bias in winter was reduced by about 0.6–3.0 °C, and the resulting modelled near-surface permafrost extent was improved (11.0–12.9 $\times 10^6$ km$^2$), comparing reasonably with observed estimates for continuous and discontinuous permafrost areas. We therefore suggest that a better-resolved snow scheme with a multi-layer snow profile should be included in next-generation reanalyses as a first step towards improving the representation of permafrost.

## 1    Introduction

Permafrost has been warming and degrading around the world (Biskaborn et al., 2019). Robust simulation of permafrost is essential for understanding responses to climate change and assessing associated changes in hydrological processes, terrain stability, and carbon losses (e.g., Westermann et al., 2016; Walter Anthony et al., 2018). The prevalence of snow cover for much of the year in permafrost regions can strongly affects soil temperature due to its influence on the surface energy balance

(Zhang, 2005; Cao et al., 2018), and is typically a key uncertainty in the representation of permafrost soil temperature (Dutra et al., 2012; Domine et al., 2019).

Climate reanalysis is a valuable source of data for permafrost science (Cao et al., 2019a). ERA5 (Hersbach et al., 2020) and the land-only reanalysis ERA5-Land (ERA5L, Muñoz-Sabater et al., 2021) are the most recent advances produced by the European Centre for Medium-Range Weather Forecasts (ECMWF) within the Copernicus Climate Change Service (C3S). The

land surface component of these reanalyses is the Tiled ECMWF Scheme for Surface Exchanges over Land with a revised land surface hydrology (HTESSEL) cycle 45r1, which is part of the ECLand modelling framework (Boussetta et al., 2021). The current HTESSEL used in ERA5 and ERA5L includes snow as an independent single layer on top of the soil layer and describes bulk temporal evolution of the snowpack (Dutra et al., 2009). In a previous study, we reported the notable warm bias





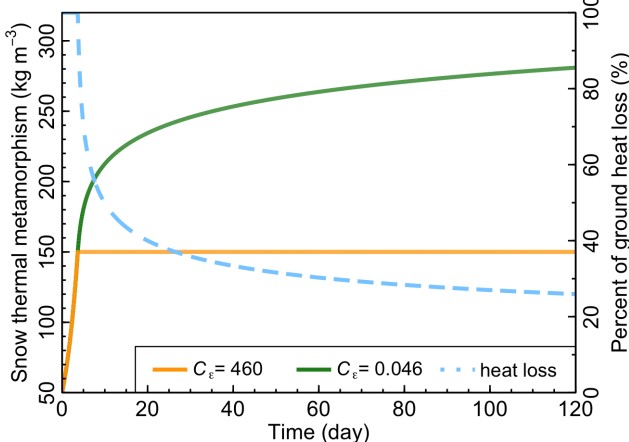

**Figure 1.** Snow compaction rate due to thermal (destructive) metamorphism (excluding liquid water) using different $c_\xi$ in Eq. (6). The dashed line is the percent of ground heat loss through the snow layer in Exp. MLS-Dis+Den compared using the $c_\xi$ of 0.046 (kg m$^{-3}$) in ERA5L, Exp. CTRL, and Exp. MLS-Std (see text for description).

of the ERA5L soil temperature in permafrost regions (Cao et al., 2020). By reviewing the snow scheme parameterization, we
determined that a low snow density in ERA5L may cause the warm bias. Through further examination of the land scheme
codes in ERA5L we identified that the current parameters do not permit thermal metamorphism to occur for snow densities
higher than 150 kg m$^{-3}$, which is far lower than metamorphosed snow in permafrost areas (Figure 1). We hypothesized that
this snow density underestimation could cause the overestimation of snow depth and hence of permafrost soil temperatures,
due to an overestimated soil-atmosphere thermal decoupling (see Cao et al., 2020).
The single-layer snow scheme (SLS) in HTESSEL can only represent the temporal evolution of snow processes at a single
time scale and lacks a solution for processes that occur at different depths and temporal scales. For this reason, a multi-layer
snow scheme (MLS) was developed independently by ECMWF (e.g., Dutra et al., 2010, 2012; Arduini et al., 2019), and is
reported to have better representation of snow physics. The MLS has added value to many other phenomena over snow-covered
regions, i.e., surface air temperature (Arduini et al., 2019). However, the state-of-the-art MLS has not been implemented in the
current scheme for the latest generation reanalysis production by ECMWF, and its impacts on soil temperature in permafrost
regions remains largely unknown.
    In this study, we introduce a new MLS recently developed by Arduini et al. (2019) into the HTESSEL, and evaluate its
capability for representing soil temperature in permafrost regions. Four simulation experiments were designed and carried out
to better understand its performance. We then evaluate the reproduced reanalysis, with a specific focus on soil temperature in
permafrost regions, by comparing temperatures with observations and original ERA5L and published permafrost products.





## 2 Snow scheme

### 2.1 ERA5-Land with single-layer snow scheme

ERA5 assimilates new datasets to improve snow representation compared to its predecessor of ERA-Interim, such as the in-situ
observations of the global Surface Synoptic Report (SYNOP) network for snow depth and snow cover information from the
Interactive Multisensor Snow and Ice Mapping System (IMS) system since 2004 (Hersbach et al., 2020). ERA5L, the land
component of ERA5, has an improved horizontal resolution of 0.1° (or ~9 km). ERA5L also uses an enhanced snow scheme
with improved snow thermal insulation compared to ERA-Interim, although it inherits the SLS (Dutra et al., 2010). ERA5L
has a soil profile of 4 layers with a total depth of 1.89 m.

### 2.2 Standard multi-layer snow scheme

The new MLS was developed and implemented in HTESSEL by Arduini et al. (2019). The MLS has a maximum of 5 snow
layers depending on the snow height ($h_{sn}$). The number of active snow layers and their thicknesses are simulated diagnostically
at the beginning of each time step before the prognostic snow fields are updated. Multiple snow layers are used when $h_{sn} > 0.1$
m, the minimum $h_{sn}$ that ensures complete snow coverage of the grid box. Over flat terrain, the depth of the uppermost snow
layer in contact with the atmosphere is fixed to 0.05 m. The second and third upper layers, and the bottom layer in contact with
the soil underneath can increase to maximum depths of 0.10 m, 0.20 m, and 0.15 m, respectively. This choice means that the
fourth layer from the top is used as an accumulation layer for deep snowpack. Take $h_{sn}$ = 1.0 m as example, the snow cover is
discretized as follows: 0.05, 0.10, 0.20, 0.50, and 0.15 m (see Arduini et al., 2019 for details).

While MLS and SLS share a set of parameters for snow densification (see Sec. 2.3), a number of physical snow processes
were changed in MLS. For example, the snow thermal conductivity ($\lambda_{sn}$, W m$^{-1}$ K$^{-1}$) in MLS is treated following Calonne
et al. (2011) and considers the influences of water vapor diffusion, $\lambda_v$, (Sun et al., 1999):

$$\lambda_{sn} = 2.5 \times 10^{-6} \rho_{sn}^2 - 1.23 \times 10^{-4} \rho_{sn} + 0.024 + \lambda_v \tag{1}$$

$$\lambda_v = (a + \frac{b}{T_{sn} - 273.16 + c}) \cdot \frac{1000}{P_a} \tag{2}$$

where $T_{sn}$ is the snow temperature (K), $P_a$ is air pressure (mb), and $a$, $b$, $c$ are calibrated values of -6.023×10$^{-2}$, -2.5425, -
289.99 from Jordan (1991). In addition, MLS includes the penetration of solar radiation within the snowpack, following Jordan
(1991).

### 2.3 Optimized multi-layer snow scheme

In this study, two parameterizations for snow discretization and densification in the MLS are optimized. A variable vertical
discretization algorithm is introduced for complex terrain, defined as regions where the standard deviation of the sub-grid scale
topography is > 50 m, following Boussetta et al. (2021). This is used to maintain a relatively high vertical resolution for snow



layers responding to fast time scales of deep snowpack development in hilly and mountainous terrain. Over complex terrain, the snow discretization is the same as in flat areas when $h_{sn} < 0.25$ m. When $h_{sn} \geqslant 0.25$ m, the minimum ($h_{sn_i}^{min}$) and maximum ($h_{sn_i}^{max}$) height for layer $i$ is variable depending on $h_{sn}$.

$$h_{sn_i}^{min} = \begin{cases} 0.25, & \Delta h_{sn} \geqslant \frac{0.15}{\alpha_0} \\ 0.10 + \alpha_0 \Delta h_{sn}, & \Delta h_{sn} < \frac{0.15}{\alpha_0} \end{cases} \tag{3}$$

where $\alpha_0$ is 0.1, and $\Delta h_{sn}$ is given as

$$\Delta h_{sn} = h_{sn} - 0.25 \tag{4}$$

$$h_{sn_i}^{max} = \begin{cases} 0.25, & i = 1 \ \& \ \Delta h_{sn} \geqslant \frac{0.15}{\alpha_0} \\ 0.10 + \alpha_0 \Delta h_{sn}, & i = 1 \ \& \ \Delta h_{sn} < \frac{0.15}{\alpha_0} \\ 0.30, & i > 1 \ \& \ \Delta h_{sn} \geqslant \frac{0.15}{\alpha_0} \\ 0.15 + \alpha_0 \Delta h_{sn}, & i > 1 \ \& \ \Delta h_{sn} < \frac{0.15}{\alpha_0} \end{cases} \tag{5}$$

Following Anderson (1976), snow densification in SLS and MLS is determined through: 1) overburden pressure, 2) thermal
metamorphism, and 3) melt metamorphism due to the presence and refreezing of liquid water in the snow layer. MLS additionally considers wind effects (snowdrift) in one-dimension by following Decharme et al. (2016). Snow density ($\rho_{sn}$, $\mathrm{kg\,m^{-3}}$) is constrained between 50 and 450 $\mathrm{kg\,m^{-3}}$. When $\rho_{sn} > 150$ $\mathrm{kg\,m^{-3}}$, the densification rate related to thermal metamorphism is parameterized as

$$\xi_{sn} = a_\xi \cdot \exp\left(-b_\xi \cdot T_D - c_\xi \cdot \Delta\beta_{sn}\right) \tag{6}$$

where the $a_\xi$ and $b_\xi$ are constant values of $2.8 \times 10^{-6}$ ($\mathrm{s^{-1}}$) and 0.042 (unitless) derived or modified from Anderson (1976) and Jordan et al. (1999). $T_D$ (K) is the temperature depression:

$$T_D = 273.16 - T_{sn} \tag{7}$$

where $T_{sn}$ is the snow temperature (K). While $c_\xi$ is empirical and highly site-specific, setting it as 460 $\mathrm{m^3\,kg^{-1}}$ in ERA5L is equivalent to halting the thermal metamorphism process for snow densities higher than 150 $\mathrm{m^3\,kg^{-1}}$ (Figure 1). This means,
for the same total snow mass, the SLS has an underestimated $\rho_{sn}$ and an overestimated $h_{sn}$, which reduces ground heat loss through the snow layer to about 26 (40) % after 120 (20) days of snowfall compared to using 0.046 $\mathrm{m^3\,kg^{-1}}$ (Figure 1a). $\Delta\beta_s$ ($\mathrm{kg\,m^{-3}}$) is given as

$$\Delta\beta_{sn} = \begin{cases} \rho_{sn} - \rho_\xi, & \rho_{sn} > \rho_\xi \\ 0, & elsewhere \end{cases} \tag{8}$$





**Table 1.** Simulation schemes including their configurations, spatial resolution, snow destructive metamorphism parameterizations ($c_\xi$) in Eq. 6, maximum number of snow layers ($N_{max}$), vertical snow discretization, and snow drift.

| Experiment | Resolution | $c_\xi$ | $N_{max}$ | Snow discretization | Snow drift |
|---|---|---|---|---|---|
| ERA5L | $0.10°$ | 460 | 1 | – | No |
| CTRL | $0.25°$ | 460 | 1 | – | No |
| MLS-Std | $0.25°$ | 460 | 5 | Arduini et al. (2019) | 1D from Decharme et al. (2016) |
| MLS-Dis | $0.25°$ | 460 | 5 | Boussetta et al. (2021) | 1D from Decharme et al. (2016) |
| MLS-Dis+Den | $0.25°$ | 0.046 | 5 | Boussetta et al. (2021) | 1D from Decharme et al. (2016) |

where $\rho_\xi$ ($kg\,m^{-3}$) is equal to $150\,kg\,m^{-3}$.

In Anderson (1976), and many snow and land surface models such as CLM (van Kampenhout et al., 2017) and Noah-MP (Yang and Niu, 2003), a value of $0.046\,m^3\,kg^{-1}$ is commonly used for $c_\xi$. Using this value, the snow thermal metamorphism was found to be more realistic (Figure 1). We therefore revised $c_\xi$ to $0.046\,m^3\,kg^{-1}$ in the optimized simulations.

## 3 Model configuration and experiment

While conducting a like-to-like evaluation facilitates comparison, running sensitivity experiments at ERA5L resolution is computationally costly and the data volume requirements are heavy. For this reason, experiments were performed using the same octahedral reduced gaussian grid as ERA5L, but with a lower horizontal spatial resolution ($\sim$28 km). All model results were then interpolated to a regular grid at a resolution of $0.25°$ based on the same interpolation method used for ERA5L. A control experiment using the same setup as ERA5L was used to assess the influence of the coarser resolution on simulated soil temperatures. Three simulation experiments with 5-layer snow schemes were completed to investigate the effect of different snow schemes and parameterizations in the HTESSEL (Table 1). A simulation with MLS as described in Arduini et al. (2019) was also performed to identify if $c_\xi$ of $460\,kg\,m^{-3}$ is problematic in soil temperature simulations, regardless of the snow scheme. Two optimized simulations, as described in Sec. 2.3, tested the capacity of optimized snow discretization and densification, respectively. All the experiments were conducted offline (Table 1).

- The control simulation (Exp. CTRL), uses the standard HTESSEL with bulk snow scheme and is conducted at a spatial resolution of $0.25°$;

- The standard MLS simulation (Exp. MLS-Std), same as CTRL but implementing the 5-layer snow scheme to HTESSEL;

- The optimized MLS simulation (Exp. MLS-Dis), same as Exp. MLS-Std but used a variable snow discretization for complex terrain;





- – The optimized MLS simulation (Exp. MLS-Dis+Den), same as Exp. MLS-Std but revised $c_\xi$ as 0.046 for snow com-
paction due to thermal (destructive) metamorphism.

## 4   Observations and evaluation

The numerical experiments were evaluated by comparing the soil temperature with a large number of observations, and
ERA5L.To streamline the comparison with previous work, we used the same observations dataset as Cao et al. (2020). The
observed soil temperatures are from various sources around the world and 639 stations in permafrost regions representing the
period of 2001–2018 (see Figure B1 from Cao et al., 2020). The dataset represents a wide range of soil temperatures, eleva-
tions, and land uses. We used BIAS (°C) to evaluate the reproduced soil temperature at a grid scale. The surface offset (SO)
in winter, as the difference between near-surface air temperature ($T_a$) and ground surface temperature (the soil temperature at
the first layer of ERA5L), describes the land-atmosphere energy exchange through the present snow layer. It is therefore also
selected here for evaluation of snow insulation effect on soil thermal regime. In the case where multiple sites were located in
the same ERA5L grid cell, BIAS was calculated for each site and then aggregated by averaging all stations in each grid cell
with equal weight. Active layer thickness was not re-evaluated here as it was minimally affected by the revised snow scheme.
Evaluation was conducted separately for different geographic regions because HTESSEL has variable performance for high-
and mid-latitudes. In addition, sites in complex terrain were used to test the suitability of optimized snow discretization.

Near-surface permafrost regions were diagnosed from the mean annual ground temperature of the fourth soil layer of repro-
duced reanalyses, and compared to the Circum-Arctic Map of Permafrost and Ground-Ice Conditions (hereafter referred to as
the IPA map, Brown et al., 1997). Given the coarse spatial resolution of reanalysis (i.e., 0.25°) and shallow soil profile, the
global land surface model like HTESSEL could only reasonably represent the presence of continuous (90–100% coverage) and
discontinuous (50–90% coverage) permafrost zones (Lawrence et al., 2008; Zhang et al., 2000). We hence apply a threshold of
50% for the permafrost zonations in the IPA map to allow for meaningful comparison with the simulated maps.

## 5   Results and discussions

### 5.1   Soil temperature

Soil temperatures in Exp. CTRL were generally close to those in ERA5L (Table 2), indicating that the experiment at a coarse-
resolution (i.e., 0.25°) is comparable to the ERA5L with a higher spatial resolution of 0.1°. Exp. MLS-Std produced slightly
warmer temperatures than Exp. CTRL because of the increased thermal decoupling between the atmosphere–snow–soil inter
faces (Dutra et al., 2012). The remaining strong warm bias in Exp. MLS-Std demonstrates that the 460 $\mathrm{m^3\,kg^{-1}}$ snow thermal
metamorphism parameter is not reasonable. The soil temperature warm bias is reduced with the optimized snow discretization
algorithm, indicating that simulation of soil temperature in complex terrain benefits from the relatively high vertical resolution
of the snow layer (Figure 2e). MLS with variable vertical discretization had a minimal influence on simulated soil temperatures
over the Tibetan Plateau. This is because the new snow discretization for complex terrain is only applied where $h_{sn}$ is $\geqslant 0.25$





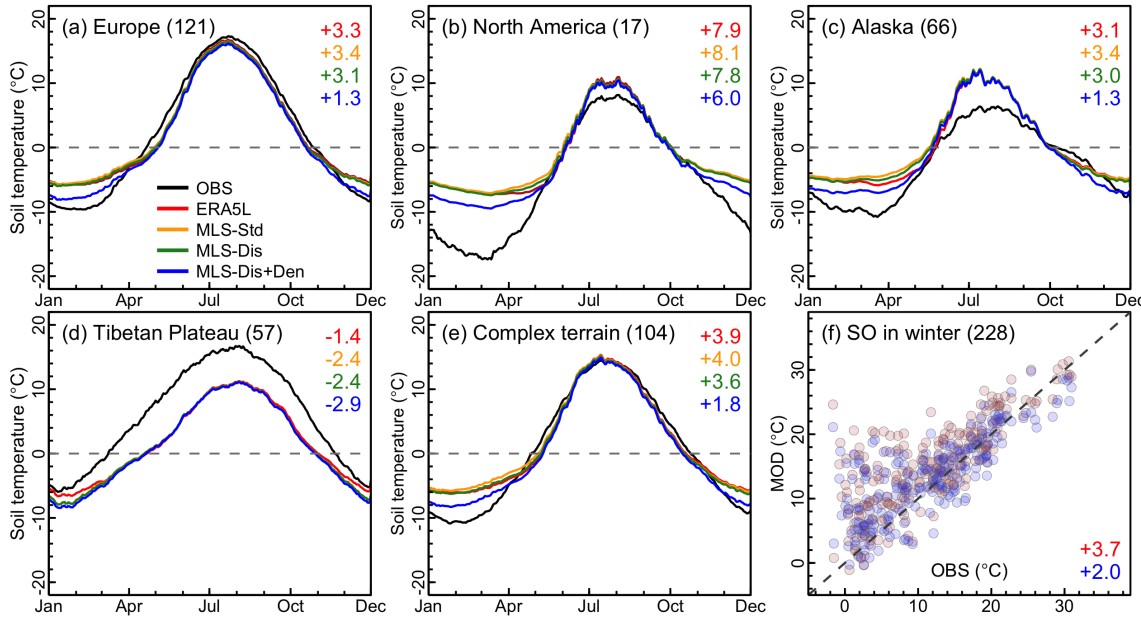

**Figure 2.** Aggregated daily soil temperature (2001–2018) for 0.07–0.28 m depth in different permafrost regions (a–e), and winter (DJF) surface offset (SO, f). The number of unique grid cells where observed sites are located is given in the bracket. Color numbers are estimated bias (°C) in winter, when snow is prevalent.

m, while snow over the Tibetan Plateau is typically very thin (Cao et al., 2019b). The Exp. MLS-Dis+Den with $c_\xi$ = 0.046 $\mathrm{m}^3\,\mathrm{kg}^{-1}$ performed best (Figure 2a, b, c), and the simulated daily soil temperature and surface offset in winter improved by about 0.6–3.0°C and 1.7°C (Table 2, Figure 2f). On the Tibetan Plateau, ERA5L soil temperature is found to have a cold bias because near-surface air temperatures are underestimated by about -5.8°C, and the new MLS enhanced this soil temperature bias.

Summer soil temperatures in all experiments are similar since the revised snow scheme mainly affects winter temperatures all have a significant warm bias in North America and Alaska. This is believed to arise due to a lack of soil organic matter representation in HTESSEL soil thermal properties (Park, 2018).

## 5.2  Near-surface permafrost extent

Exp. MLS-Std underestimated near-surface permafrost extent compared to the extent of continuous and discontinuous per-
mafrost area on the IPA map (11.8–14.6 $\times 10^6$ km$^2$, Brown et al., 1997), due to the overestimated soil temperature (Figure 3). The estimated global near-surface permafrost area increases from 8.8–10.0 $\times 10^6$ km$^2$ in ERA5L to 11.0–12.9 $\times 10^6$ km$^2$ in the Exp. MLS-Dis+Den scheme during 2001–2018, with the increase primarily represented along the southern fringes of permafrost in eastern Siberia and Canada. This is more reasonable when comparing to the observed distribution on the IPA map. The improved soil temperature in Exp. MLS-Dis+Den indicated that near-surface permafrost area decreased at a rate of

**Table 2.** BIAS of near-surface air temperature ($T_a$), soil temperature ($T_s$) for four layers in DJF at a grid scale, and permafrost area (PA) estimated from the mean annual ground temperature of the fourth soil layer (1.89 m).

| Experiment | Europe | | North America | | Alaska | | Tibetan Plateau | | Complex Terrain | | PA |
|---|---|---|---|---|---|---|---|---|---|---|---|
| | $T_a$ | $T_s$ | $T_a$ | $T_s$ | $T_a$ | $T_s$ | $T_a$ | $T_s$ | $T_a$ | $T_s$ | |
| | (°C) | (°C) | (°C) | (°C) | (°C) | (°C) | (°C) | (°C) | (°C) | (°C) | ($10^6$ km²) |
| ERA5L | 1.5 | 0.9 to 6.0 | -0.6 | 4.9 to 9.3 | -2.1 | 1.1 to 3.1 | -5.7 | -2.7 to -1.4 | 0.3 | 1.4–3.9 | 8.8–10.6 |
| CTRL | 1.4 | 0.9 to 6.0 | -0.8 | 4.0 to 8.5 | -2.2 | 0.9 to 3.0 | -5.8 | -2.6 to -1.6 | 0.1 | 1.2–3.7 | 8.8–10.5 |
| MLS-Std | 1.5 | 1.0 to 5.6 | -0.8 | 5.2 to 9.5 | -2.2 | 1.1 to 3.4 | -5.8 | -2.9 to -2.2 | 0.2 | 1.4–4.0 | 7.8–9.5 |
| MLS-Dis | 1.5 | 0.8 to 5.3 | -0.9 | 4.9 to 9.3 | -2.2 | 0.8 to 3.0 | -5.8 | -2.9 to -2.2 | 0.2 | 1.1–3.6 | 8.0–10.0 |
| MLS-Dis+Den | 1.5 | -0.7 to 3.0 | -0.8 | 3.7 to 7.6 | -2.2 | -0.6 to 1.3 | -5.8 | -3.3 to -2.7 | 0.2 | -0.3–1.8 | 11.0–12.9 |

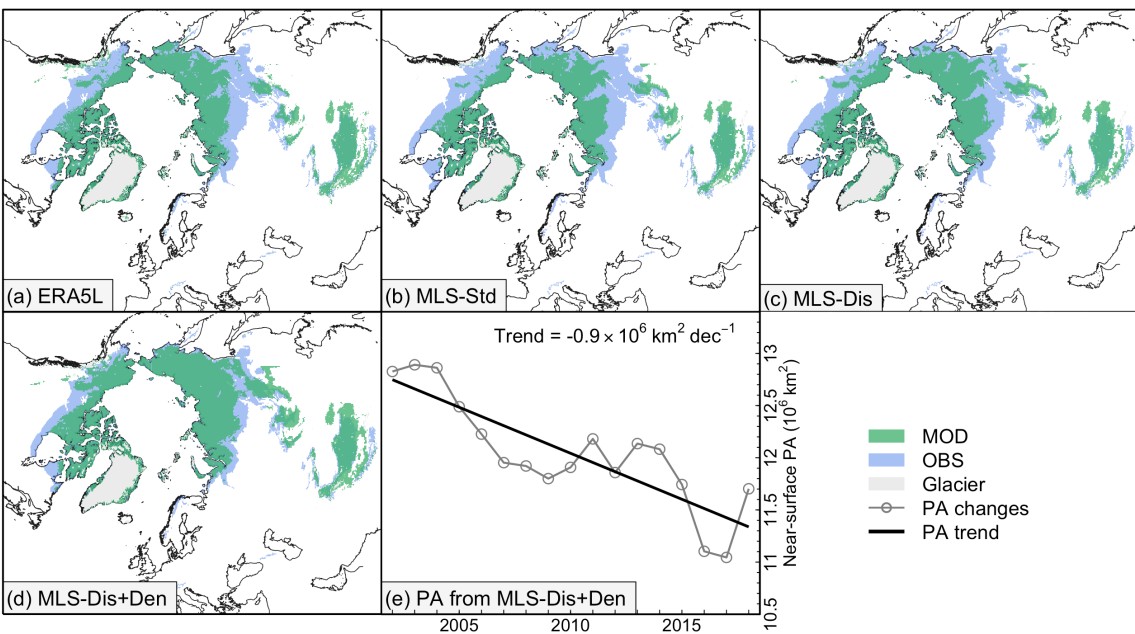

**Figure 3.** Observed estimates for continuous and discontinuous permafrost area (PA) from the IPA map (OBS) and near-surface permafrost area derived from HTESSEL with different model schemes (MOD) in 2001–2002. The near-surface permafrost area trend (e) is derived from the Exp. MLS-Dis+Den during 2001–2018.

$0.9 \times 10^6$ km² dec$^{-1}$, corresponding to a loss of $1.41 \times 10^6$ km² since 2002. This is similar to previous land surface model simulations, i.e., $0.9$–$1.1 \times 10^6$ km² dec$^{-1}$ from 1990–2040 in Lawrence et al. (2008).

## 6 Conclusion

We identified that a bias toward lower snow density arising from an unreasonable thermal (destructive) metamorphism parameterization in snow densification routines is one of the main sources for a warm bias in ERA5L soil temperature in permafrost
regions, particularly in high latitude areas. We implemented and evaluated a recently developed MLS for ECLand, with revised/optimized parameters for snow thermal metamorphism. Using an optimized version of the multi-layer snow scheme in HTESSEL significantly improved winter daily soil temperature simulation in high-latitude permafrost regions by about 0.6–3.0°C, and produced a better representation of permafrost extent. Since most current reanalyses use the single-layer snow scheme, we suggest that a better-resolved snow scheme with a multi-layer snow profile should be included in next-generation
reanalysis as a first step towards improving representation of permafrost conditions.

The numerical experiments were conducted offline with no coupling of the land surface to the atmospheric fields. Therefore, the influence of the revised snow scheme on the atmosphere is not considered, and further online simulations that include coupling with atmospheric processes is suggested for future study.

*Data availability.* Temperature observations are access from Cao et al. (2020). The improved soil temperature along with snow variables in
numerical experiment MLS-Dis+Den is produced at a daily resolution and is public open and via Zenodo (10.5281/zenodo.6008390).

*Author contributions.* BC proposed the initial idea and carried out this study by designing simulation experiments, analyzing data, organizing, and writing the paper and was responsible for the compilation and quality control of the observations. GA developed and implemented the multi-layer snow scheme in HTESSEL, conducted the experiment, and reproduced reanalysis. All authors contributed to the writing of the paper.

*Competing interests.* The authors declare that they have no conflict of interest.

*Acknowledgements.* The authors thank Stephan Gruber and Baohong Ding for the helpful discussion and comments. ERA5 reanalysis data is provided by the ECMWF. BC was supported by the National Natural Science Foundation of China (NSFC) (grant no. 41988101, 42101134).



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
