# Peer review of "Brief communication: Improving ERA5-Land soil temperature in permafrost regions using an optimized multi-layer snow scheme"

_The Cryosphere, 2022_

## Author Comment (AC1)

**Author's Responses to RC1's comments on "Brief communication: Improving ERA5-Land soil temperature in permafrost regions using an optimized multi-layer snow scheme"**

Bin Cao1, Gabriele Arduini2, Ervin Zsoter2,3

1National Tibetan Plateau Data Center, Institute of Tibetan Plateau Research, Chinese Academy of Sciences, Beijing, China

2European Centre for Medium-Range Weather Forecasts, Reading, UK

3Department of Geography and Environmental Science, University of Reading, Reading, UK

Correspondence: Bin Cao (bin.cao@itpcas.ac.cn)

The authors would like to thank the reviewer for constructive feedback, and the thorough assessment of the manuscript. Below we provide a point-to-point response to each comment, reviewer comments are given in black, responses are given in blue. Additionally, we have included details of how we intend to address these changes in a revised submission.

This short communication is easy to read and follow. The conclusion is very clear and also important. Generally, I am willing to recommend its publication after the following comments are considered.

• For the evaluation in Figure 2, the improvement seems to fail on the Tibetan Plateau. Except for very thin snow as a reason, perhaps the comparison itself has an impact? The comparison is based on site (OBS) vs. grid (SIM) scale? if so, such comparison may have partial impacts because of very complex terrain of the Tibetan Plateau. Response: We agree scale gap could be a possible reason, especially over the Tibetan Plateau. The worse performance in revised simulation is thought to because of underestimated near-surface air temperature rather than snow scheme. The near-surface air temperature over the TP is significantly underestimated by about -5.8  $\pm$  3.7°C in winter (DJF, see Table 2). This generally could account the underestimated soil temperature, i.e., from -3.3 to -2.7°C.

In addition, Orsolini et al. (2019) reported that excessive snowfall might be the primary factor for the large overestimation of snow depth and cover in ERA5(-land) reanalysis. In Sec 5.1, we intend to revise as below to clarify.

"On the Tibetan Plateau, the soil temperature in Exp. MLS-Dis+Den is found to have a worse performance compared to ERA5L. This is because  $T_a$  is significantly underestimated by about -5.8  $\pm$  3.7°C over the Tibetan Plateau, which could account for the cold bias of soil temperature, i.e., from -3.3 to -2.7°C (Table 2). While the new MLS reduced overestimated snow depth (Figure 21), it suppressed snow insulation and hence enhanced soil temperature cold bias. Previous studies indicated that excessive winter precipitation in ERAL(-Land) might be an additional uncertainty for the remarkable overestimation of snow depth over the TP Orsolini et al. (2019)."

In addition, we indented to reformulated Figure 2 in order to give detailed time-series and spatial information of soil temperature bias, as suggested by the editor.

• I think that the diagnostic method for near-surface permafrost needs to be introduced more detailed. For instance, "permafrost is identified as ground where monthly soil temperature is less than 0°C for 24 consecutive months in at least one layer of the simulated upper 4 soil layers", as the statements from (Guo and Wang, 2017, https://doi.org/10.1002/2017JD027691).

Response: In the second paragraph of Sec.4, we intend to revise as below to clarify.

"Near-surface permafrost regions were diagnosed from the mean annual ground temperature of the fourth soil layer of reproduced reanalyses, i.e., soil temperature is less than 0°C for two consecutive years, and is compared to the Circum-Arctic Map of Permafrost and Ground-Ice Conditions (hereafter referred to as the IPA map, Brown et al., 1997)".

• For the evaluation in Figure 3, I think that the authors should add more discussions on why the simulated permafrost extent is still smaller than the IPA map. For instance, different periods for generating the simulation and IPA map; the simulation is only the results at 0~1.9 m depth, may be different from the IPA map.

---

## Author Response (AR1)

**Author's Responses to the comments on *"Brief communication: Improving ERA5-Land soil temperature in permafrost regions using an optimized multi-layer snow scheme"**

Bin Cao[1], Gabriele Arduini[2], Ervin Zsoter[2,3]

[1]National Tibetan Plateau Data Center, Institute of Tibetan Plateau Research, Chinese Academy of Sciences, Beijing, China
[2]European Centre for Medium-Range Weather Forecasts, Reading, UK
[3]Department of Geography and Environmental Science, University of Reading, Reading, UK

**Correspondence**: Bin Cao (bin.cao@itpcas.ac.cn)

The authors would like to thank the editor and reviewer for constructive feedback, and the thorough assessment of the manuscript. Below we provide a point-to-point response to each comment, editor and reviewer comments are given in black, responses are given in blue. Additionally, we have included details of how we addressed these changes in the revised submission.

**Editor**

Overall, the analysis methodology is clear, and carefully applied. The sequence of model configurations is logical and clearly described. I have some concerns regarding the approach to aggregate the performance metrics over large regions and long time periods as illustrated in Figure 2. While the improvement in overall bias is apparent, it's not possible to diagnose much insight into the drivers of this improvement. Changes to this analysis do not need to be made at this stage, but I encourage the authors to consider whether a less generalized (in space and time) diagnosis of the simulations is possible. At the very least, it would be instructive to see how the snow profile varies for the four experimental configurations at a validation site with snow and soil measurements. I know this type of detailed evaluation was provided in Arduini et al. (2019) but it would be good to know how the obtained bias improvements correspond to changes in the simulated snow stratigraphy. Please consider this comment along with the reviewer input when it is received later in the process.
Response: We conducted a brief snow depth evaluation and the spatial soil temperature bias is added in Figure 2 as also suggested by RC2. Since the spatial picture of snow depth and soil temperature performance are available in Figure 2 now, we argue the time-series at specific sites are not really necessary. Instead, the time-series of four selected stations from different regions are shown in the supporting information.

**Reviewer 1**

This short communication is easy to read and follow. The conclusion is very clear and also important. Generally, I am willing to recommend its publication after the following comments are considered.

- For the evaluation in Figure 2, the improvement seems to fail on the Tibetan Plateau. Except for very thin snow as a reason, perhaps the comparison itself has an impact? The comparison is based on site (OBS) vs. grid (SIM) scale? if so, such comparison may have partial impacts because of very complex terrain of the Tibetan Plateau.
Response: We agree scale gap could be a possible reason, especially over the Tibetan Plateau, and this requires a comprehensive evaluation with dense measurements. The worse performance in revised simulation is thought to because of underestimated near-surface air temperature rather than snow scheme. The near-surface air temperature over the TP is significantly underestimated by about -5.8 $\pm$ 3.7°C in winter (DJF, see Table 2). This generally could account the underestimated soil temperature, i.e., from -3.3 to -2.7°C.
In addition, Orsolini et al. (2019) reported that excessive snowfall might be the primary factor for the large overestimation of snow depth and cover in ERA5(-land) reanalysis. In Sec 5.1, we revised as below to clarify.

  *"On the Tibetan Plateau, the soil temperature in Exp. MLS-Dis+Den is found to have a worse performance compared to ERA5L. This is because $T_a$ is significantly underestimated by about -5.8 $\pm$ 3.7°C over the Tibetan Plateau, which could account for the cold bias of soil temperature, i.e., from -3.3 to -2.7°C (Table 2). While the*

*new MLS reduced overestimated snow depth (Figure 2I), it suppressed snow insulation and hence enhanced soil temperature cold bias."*

*"Orsolini et al. (2019) revealed that excessive winter precipitation in ERAL(-Land) might be an additional uncertainty for the remarkable overestimation of snow depth over the Tibetan Plateau."*

In addition, we indented to reformulated Figure 2 in order to give detailed time-series and spatial information of soil temperature bias, as suggested by the editor.

- I think that the diagnostic method for near-surface permafrost needs to be introduced more detailed. For instance, "permafrost is identified as ground where monthly soil temperature is less than 0°C for 24 consecutive months in at least one layer of the simulated upper 4 soil layers", as the statements from (Guo and Wang, 2017, https://doi.org/10.1002/2017JD027691).
  Response: In the second paragraph of Sec.4, we revised as below to clarify.

  *"Regions with the presence of near-surface permafrost were diagnosed based on the mean annual ground temperature of the fourth soil layer of reproduced reanalyses, i.e., where soil temperature is less than 0 °C for two consecutive years, and compared to the Circum-Arctic Map of Permafrost and Ground-Ice Conditions (hereafter referred to as the IPA map, Brown et al., 1997)".*

- For the evaluation in Figure 3, I think that the authors should add more discussions on why the simulated permafrost extent is still smaller than the IPA map. For instance, different periods for generating the simulation and IPA map; the simulation is only the results at 0∼1.9 m depth, may be different from the IPA map.
  Response: In section 5.2, we added the following sentences to clarify.

  *"Besides to the model uncertainties, such as the shallow soil profile, the smaller simulated permafrost area compared to the IPA map could be traced to the different periods represented, i.e., a few decades prior to 1990 for the IPA map and 2001–2018 for ERA5L. Furthermore, because permafrost is a hidden phenomenon, its extent is fundamentally difficult to be observed and validation."*

- For simulation experiment, the period for spin-up (initiation) should be described. Because soil temperature are analyzed in this study and if they reach a stability in the simulation. In my opinion, a recent study shows that this is important for permafrost simulation, especially for the Tibetan Plateau.
  Response: Xue et al. (2021) recently reported the possible influences of initialized land surface temperature on seasonal simulations, and we agree unsuitable spin-up would result in significant uncertainties.
  The offline simulation experiments in this study were all initialised from ERA5 on 1 January 1979. Therefore a 20-year spin-up period is considered before analysing the data. We added the model initialization and spin-up period in Sec.3 Model configuration and experiment.

  *"The offline simulation experiments in this study were all initialised from ERA5 on 1 January 1979, and the period of 1979–2000 is used to spin up before simulation and analyses were conducted."*

**Reviewer 2**

I have produced this review 'blind' and have not looked at the comments posted by the authors that respond to the first reviewer. So apologies if there is any repetition in the points I make.
Response: Thanks for the independent comments.

This paper presents the (hitherto unknown) impacts of the implementation of a new multi-layer snow (MLS) scheme into the land surface component of ERA5 reanalysis (HTESSEL) which was currently operating with a single-layer snow scheme. ERA5 outputs are widely used by the community so a critical appraisal of it's ability to simulate permafrost regions is welcomed and necessary. One of the limitations of the current model set up is that thermal snow metamorphism is programmed to not respond to thermal forcing above densities of 150kg/m3 therefore the simulated densities are artificially low and because of thermal conductivity of snow, ground temperatures biased high. A sensible suite of comparative experiments were proposed to demonstrate how each incremental adjustment to the snow scheme affected soil temperature and led to moderate improvements in some regions (Europe, Alaska and complex terrain) but little/no improvement in others (North America/Tibet) – which is very interesting. Inevitably, the representation of snow and it's layer properties will not be accurate – we see this in most/all major land surface models but addition of a multi-layer scheme is a good step forward.

The average temperature biases are still quite large over some regions so this should be reported in more detail. For this reason I think Figure 2 could be better used to illustrate exactly where the world ERA5 gets subsurface temperatures wrong/right. Please see my major comments.

Response: We agree spatial information of revised simulations are missing in the current format. The handling editor also reported similar concern in the previous review before publication in TCD. We revised Figure 2 to add the spatial performance of HTESSEL with a MLS in snow and soil temperature representations (see below).

The snow depth was improved at most sites in MLS (Figure 2A–C). For the Tibetan Plateau, a remarkable uncertainty is the atmospheric forcing. The daily near-surface air temperature is significantly underestimated by about $5.8 \pm 3.7\,°C$. This means a perfect snow and soil schemes will lead to cold biased soil temperatures. The Exp. MLS-Dis+Den simulation slightly reduced snow depth bias (see Figure Snow in TP) over the TP but leads to a even underestimated soil temperature due to the stronger snow thermal conductivity. The snow depth overestimation could be traced back to the overestimated snowfall (Orsolini et al., 2019). In sec. 5.1 of the revised submission, we changed as below to clarify.

*"On the Tibetan Plateau, the soil temperature in Exp. MLS-Dis+Den is found to perform worse than ERA5L. This is because $T_a$ is significantly underestimated by about -5.8 ± 3.7°C over the Tibetan Plateau, which could account for the cold bias of soil temperature, i.e., from -3.3 to -2.7°C (Table 2). While the new MLS reduced overestimated snow depth (Figure 2B), it suppressed snow insulation and hence enhanced soil temperature cold bias."*

*"Orsolini et al. (2019) revealed that excessive winter precipitation in ERAL(-Land) might be an additional uncertainty for the remarkable overestimation of snow depth over the Tibetan Plateau."*

We agree the warm bias of soil temperature in North America and Alaska (both in summer and winter) were still remarkable, although the new MLS reduced the cold bias by about 1.9/1.8 °C in North America/Alaska. This was thought to raise from lacking a soil organic representation in the soil column, and we revised the discussion of soil organic matter (in sec 5.1) as below to clarify.

*"Summer soil temperatures in all experiments are similar since the revised snow scheme mainly affects winter temperatures. In North America and Alaska, soil temperatures generally have a significant warm bias (Figure 2). This is thought to be related to a lack of vertical variation in soil texture within the soil column in HTESSEL, which would allow a more sophisticated treatment of soil organic matter and its impact on soil thermal properties (Park, 2018)."*

A detailed evaluation and improvement of the MLS requires stand-alone simulations that forced by in situ atmospheric observations.

Overall this is a sound paper which makes a first attempt to improve soil temperature reconstructions and is appropriate for a Brief Communication format but I think at least one Figure could be revised in order to squeeze some more detail out of the analyses provided. Please see my major/minor comments.

**Major:**

- **More detail:** I feel interested readers would definitely like to see spatial patterns of temperature biases, rather than summary figures as are presented here so that Fig. 3 (permafrost map) may be put into better context about what is driving the discrepancy(ies). The high levels of aggregation (over time, depth and space) are likely masking some very interesting features of this output. The under representation of permafrost regions seems to be related to soil temperatures being too warm in Winter in all cases. Suggestion: Why not show a 4 (region) x 4 (model experiment)-panelled figure of maps of 0.07-0.28m DJF Bias for the Northern Hemisphere so readers can see what the spatial patterns of this discrepancy looks like ?
  Response: We agree the manuscript will significantly benefit from a spatial patterns of soil temperature biases. Based on the requirements of TC, the brief communication should have a maximum of 3 figures and/or tables. The manuscript in its current format is already oversized with 3 figure and 2 tables. In addition, we need to make a balance of snow and soil temperature in the manuscript. For these reasons, we added the spatial snow depth and soil temperature bias into Figure 2 rather than present in a separate one. In addition, we added a brief clarification of snow depth performance in Sec. 5.1 (see below).

  *"Snow depth is generally improved at most observed sites (Figure 2A and B), and the overestimation in ERA5L was reduced from 0.19 m to 0.08–0.11 m in the MLS (Figure 2C). However, there is little improvement (i.e, 3 cm) over the Tibetan Plateau. Orsolini et al. (2019) revealed that excessive winter precipitation in ERAL(-Land) might be an additional uncertainty for the remarkable overestimation of snow depth over the Tibetan Plateau."*

- **Thermal Conductivity:** Although I am happy Calonne is an appropriate choice for thermal conductivity parameterisation I'd like to ask the authors for clarification about including the water vapour term which isn't included in Calonne's original equation. We know water vapour diffusion affects thermal conductivity and is difficult to represent. However, Calonne developed their equation based on a quadratic line of best fit using observations of snow density and thermal conductivity. Although the equation is presented solely as a function

of density, it is highly probable that when the snow and density measurements featured in Calonne's Fig. 1 were taken that there might have been some water vapour diffusion happening so this effect could (or maybe not) already be implicitly included in the equation by Calonne. My concern is that the addition of the water vapour term by the authors may cause this effect to be 'double counted'. I'd recommend the authors re-run one of the experiments (e.g. MLS Dis+Den since this provides the biggest improvement) without the additional water vapour diffusion term in the Calonne equation and report back on whether omission of this term causes a significant change in the results, or not.

Response: We agree the water vapor has significant influences on snow thermal conductivity, and treating in different parameterizations would affect simulated soil temperature (Figure Thermal conductivity). However, the parameterization present by Calonne et al. (2011) considered "purely conductive effects (i.e., conduction through ice and interstitial air)·" and non-conductive processes (i.e., water vapor diffusion and air convection) was separately (see paragraph 21 from Calonne et al. (2011)). Calonne et al. (2011) also clarified in the Abstract and we simply copied the text here.

*"Only conduction through ice and interstitial air were considered. The obtained values are strongly correlated to snow density."*

In addition, Fourteau et al. (2020) reported that the method proposed by Calonne et al. (2011) was *"treating heat conduction as decoupled from vapor transport"* (please see the last sentence in paragraph 2 from Fourteau et al. (2020)).

In such a case, involving an additional water vapor diffusion in snow effective thermal conductivity is reasonable. We therefore donot think the manuscript will benefit from a additional simulation experiment without considering the water vapor of thermal conductivity. Instead, we revised as below to clarify.

*"For example, the effects of heat conduction and water vapor ($\lambda_v$) on snow thermal conductivity ($\lambda_{sn}$, $W\,m^{-1}\,K^{-1}$) is treated separately in MLS. The former was parameterized following Calonne et al. (2011), and the later was calculated using equation from Sun et al. (1999)."*.

- **Model spin up:** As an experienced user of CESM2.0, which I know is not the model considered here, we've found that subsurface temperatures are extremely sensitive to spin up procedure. The authors should include information about their spin up procedure and whether they have tested the sensitivity of the results to it. See next point......

Response: RC1 had the similar comments. The offline simulation experiments in this study were all initialised from ERA5 on 1 January 1979. Therefore a 21-year spin-up period is considered before analysing the data (2001–2018). We added the model initialization and spin-up period in Sec.3 Model configuration and experiment (see below).

*"The offline simulation experiments in this study were all initialised from ERA5 on 1 January 1979, and the period of 1979–2000 is used to spin up before simulation and analyses were conducted."*

- **Permafrost extent:** The revisions don't manage to capture southern extent at all but you are only considering 2001-2002. Could this just be an anomalous year to compare to ? Or could this be a result of the spin up procedure contributing to the model state being artificially too warm to start with ? I'm not sure you should read too much into representation of a single year. Perhaps you could add some extra lines to map to show annual extent of the permafrost region as predicted by the model over the time period considered so that the reader can understand the variability of the predicted extent over the time period considered ?
Response: The simulated permafrost region is generally smaller than "observed" estimate, and hence could not "capture" the permafrost southern boundary. The 21-years spin-up for soil temperature within 1.89 m is robust (please see our responses to your previous comment).

The underestimated permafrost region could be raised from many reasons, in this study, we proposed three aspects:

[1] Period gap: Our simulation covers the period of 2001–2018 while IPA map represents permafrost condition in a few decades prior to 1990. We hence could expect simulated permafrost area here is lower than the

IPA map due to climate warming.

[2] Model uncertainties: Given the fact that permafrost could be present several meters beneath the ground surface even in continuous and discontinuous permafrost zones, current HTESSEL or ECLand model with a very shallow soil profile (i.e., 1.89 m) is not sufficient for deep permafrost simulations. In this context, the simulated permafrost here remains as "near-surface permafrost", and hence could be smaller than the IPA map.

[3] Fundamental challenge: Because permafrost extent is a variable that cannot be observed, we fundamentally lack possibilities for proper validation (Gruber, 2012). The observed permafrost extent, i.e., IPA map, is subjected to significant uncertainties although it was widely used (Cao et al., 2019).

In section 5.2, we added the following sentences to clarify.

*"Besides model uncertainties, such as the shallow soil profile, the smaller simulated permafrost area compared to the IPA map could be traced to the different periods represented, i.e., a few decades prior to 1990 for the IPA map and 2001–2018 for ERA5L and the simulation experiments. Furthermore, because permafrost is a hidden phenomenon, its extent is fundamentally difficult to be observed and validated."*

- **Multi- layer snow scheme:** – Perhaps the authors could comment on how well (or not) the scheme simulates the different types of snow we see across the northern hemisphere. It's good getting density and thermal conductivity right but if the snow layering and density is not accurate (which I suspect it likely is) this will feed into erroneous thermal conductivity and soil temperatures. Just a comment or acknowledgement required.
Response: We revised the Figure 2 (also see our responses to your major comment 1). In Sec. 5.1 Soil temperature, a brief comment on snow depth performances was added. However, We believe that an evaluation of snow density and snow internal properties is beyond the scope of this work.

*"Snow depth is generally improved at most observed sites (Figure 2A and B), and the overestimation in ERA5L was reduced from 0.19 m to 0.08–0.11 m in the MLS (Figure 2C). However, there is little improvement (i.e, 0.03 m) over the Tibetan Plateau. Orsolini et al. (2019) revealed that excessive winter precipitation in ERAL(-Land) might be an additional uncertainty for the remarkable overestimation of snow depth over the Tibetan Plateau."*

**Minor:**

- Figure 1: Typo: Greek symbols differ between being Cepsilon in the legend and Cxi in the figure caption. I think you need to change epsilon?
Response: Thanks! In the Figure legend, it should be changed to $c_\xi$. In the revised submission, we changed to figure 1 as below.

- Figure 2 (f) Do the blue and red colours refer to ERA5L and MLS-Dis+Den ? If so please add some text to caption for clarity. But please consider revising this figure to show more geographical detail in the biases as per my major comment.
Response: Yes, red means to ERA5L and blue for Exp. MLS-Dis+Den. The legend in subfig a is applicable for all figures. We revised the figure with spatial soil temperature bias, and the legend text was moved to the bottom right (please see the revised Figure 2 at the end of the response.).

- Line 6: Over which region ?
Response: At the observed stations in high-latitude permafrost regions. We changed as below in the revised submission.

*"The daily warm bias in winter was reduced by about 0.6–3.0 °C at the 522 observed stations in high-latitude permafrost regions."*

- Line 7: Be specific – for which time period are the permafrost area simulations relevant for ?
Response: Revised to *"...11.0–12.9 $\times 10^6$ km$^2$ during 2001–2018"*....

- Line 65 – These values were derived by Sun et al. themselves, it appears, not Jordan according to the Sun et al. 1999 paper – See their Appendix A.
Response: Revised to the corrected reference of Sun et al. (1999).

- Line 68: What does 'optimized' mean in this context ? When I think of optimization I think of tuning parameters to replicate observations, which doesn't appear to be the case here. The following sentences just simply describe two techniques the introduce sperate densification and layering for differing terrains.
  Response: The tile of Sec 2.3 is changed as "Revised multi-layer snow scheme" in a revised submission.

- Line 121: Land type, or vegetation coverage can have significant effects on the way that snow models evolve the physical properties of the snow. Presenting the data in Figure 2 as maps of biases may give clues as to whether this is a factor.
  Response: Spatial bias of snow depth and soil temperature is added in Figure 2 in the revised submission.

- Line 121: Why is the word 'BIAS' capitalised ? Is it a software or do you simply mean 'bias' ?
  Response: Yes, we meant bias. The *'BIAS'* is changed to *'bias'* throughout the manuscript.

- Line 126-127: Please expand – variable performance in which aspects of climate simulation ?
  Response: We meant the simulations conducted here will be affected by the model forcing or the ERA5 atmospheric component, which has variable performance in different geographic regions. We revised as below.

  *"Evaluation was conducted separately for different geographic regions because the atmospheric component of ERA5, used as HTESSEL forcing, has variable performance for high- and mid-latitudes."*

- Line 143: 'MLS with' – should this be changed to : 'A MLS with'
  Response: It is revised as suggested.

- Line 147: I do not see this range (0.6–3.0° and 1.7°C) reflected in the Table or figure. You say it is an improvement – but improvement with respect to what, and for which regions ? Lines 146-147 are not clear to me.
  Response: The numbers are reduced wBias of aggregated daily soil temperature in high-latitude permafrost regions in Exp. MLS-Dis+Den comparing to ERA5L. We revised this part as below to clarify.

  *"The Exp. MLS-Dis+Den with $c_\xi$ = 0.046 $m^3$ $kg^{-1}$ performed best among the simulations (Figure 2G–I). Comparing to ERA5L, the Exp. MLS-Dis+Den reduced winter wBias of aggregated daily soil temperature and surface offset by about 0.6–3.0 °C (Figuree 2G–J) and 1.7 ° (Figure e 2K) at the observed sites in high-latitude permafrost regions (Table 2)."*

- Line 152: ...affecting soil thermal conductivity ? This could be another inaccuracy in HTESSEL
  Response: Mostly thermal conductivity but also thermal capacity (that's why we used *"thermal properties"*). Lacking a representation of soil organic matter is thought be a major uncertainties in soil temperature simulations, and the discussion is added as below.

  *"Summer soil temperatures in all experiments are similar since the revised snow scheme mainly affects winter temperatures. In North America and Alaska, soil temperature are generally have a significant warm bias year round (Figure 2H and I). This is thought to be related to a lack of vertical variations of soil textures within the soil column in HTESSEL, which would allow a more sophisticated treatment of soil organic matter and its impact on soil thermal properties (Park, 2018)."*.

- Line 161: it isn't appropriate to recent permafrost loss rates (2002-present) to future projections to 2040.
  Response: The permafrost area changing rate during 1990–2040 from Lawrence et al. (2008) is generally linear (see Figure 5 from Lawrence et al. (2008)), and hence comparable.

- Line 174: "Temperature observations are access from Cao et al. (2020)" − > rephrase − > "Temperature observations were made available by the authors of Cao et al. (2020)."
  Response: Changed to *"Temperature observations were made available by the authors of Cao et al. (2020)."* in a revised submission.

**References**

Arduini, G., Balsamo, G., Dutra, E., Day, J. J., Sandu, I., Boussetta, S., and Haiden, T.: Impact of a Multi-Layer Snow Scheme on Near-Surface Weather Forecasts, Journal of Advances in Modeling Earth Systems, 11, 4687–4710, https://doi.org/10.1029/2019MS001725, 2019.

Brown, J., Ferrians, O., Heginbottom, J., and Melnikov, E.: Circum-Arctic map of permafrost and ground-ice conditions, Circum-pacific map series CP-45, scale 1:10,000,000, Tech. rep., U.S. Geological Survey in Cooperation with the Circum-Pacific Council for Energy and Mineral Resources, Washington, DC, 1997.

Calonne, N., Flin, F., Morin, S., Lesaffre, B., Du Roscoat, S. R., and Geindreau, C.: Numerical and experimental investigations of the effective thermal conductivity of snow, Geophysical Research Letters, 38, 1–6, https://doi.org/10.1029/2011GL049234, 2011.

Cao, B., Zhang, T., Wu, Q., Sheng, Y., Zhao, L., and Zou, D.: Brief communication: Evaluation and inter-comparisons of Qinghai–Tibet Plateau permafrost maps based on a new inventory of field evidence, The Cryosphere, 13, 511–519, https://doi.org/10.5194/tc-13-511-2019, 2019.

Fourteau, K., Domine, F., and Hagenmuller, P.: Impact of water vapor diffusion and latent heat on the effective thermal conductivity of snow, The Cryosphere Discussions, pp. 1–25, https://doi.org/10.5194/tc-2020-317, 2020.

Gruber, S.: Derivation and analysis of a high-resolution estimate of global permafrost zonation, The Cryosphere, 6, 221–233, https://doi.org/10.5194/tc-6-221-2012, 2012.

Lawrence, D. M., Slater, A. G., Romanovsky, V. E., and Nicolsky, D. J.: Sensitivity of a model projection of near-surface permafrost degradation to soil column depth and representation of soil organic matter, Journal of Geophysical Research, 113, F02 011, https://doi.org/10.1029/2007JF000883, 2008.

Orsolini, Y., Wegmann, M., Dutra, E., Liu, B., Balsamo, G., Yang, K., de Rosnay, P., Zhu, C., Wang, W., Senan, R., and Arduini, G.: Evaluation of snow depth and snow cover over the Tibetan Plateau in global reanalyses using in situ and satellite remote sensing observations, The Cryosphere, 13, 2221–2239, https://doi.org/10.5194/tc-13-2221-2019, 2019.

Park, S.: IFS doc-Physical processes, pp. 1–223, 2018.

Sun, S., Jin, J., and Xue, Y.: A simple snow-atmosphere-soil transfer model, Journal of Geophysical Research: Atmospheres, 104, 19 587–19 597, https://doi.org/10.1029/1999JD900305, 1999.

Xue, Y., Yao, T., Boone, A., Diallo, I., Liu, Y., Zeng, X., Lau, W., Sugimoto, S., Tang, Q., Pan, X., van Oevelen, P., Klocke, D., Koo, M.-S., Lin, Z., Takaya, Y., Sato, T., Ardilouze, C., Saha, S., Zhao, M., Liang, X.-Z., Vitart, F., Li, X., Zhao, P., Neelin, D., Guo, W., Yu, M., Qian, Y., Shen, S., Zhang, Y., Yang, K., Leung, R., Yang, J., Qiu, Y., Brunke, M., Chou, S. C., Ek, M., Fan, T., Guan, H., Lin, H., Liang, S., Materia, S., Nakamura, T., Qi, X., Senan, R., Shi, C., Wang, H., Wei, H., Xie, S., Xu, H., Zhang, H., Zhan, Y., Li, W., Shi, X., Nobre, P., Qin, Y., Dozier, J., Ferguson, C., Balsamo, G., Bao, Q., Feng, J., Hong, J., Hong, S., Huang, H., Ji, D., Ji, Z., Kang, S., Lin, Y., Liu, W., Muncaster, R., Pan, Y., Peano, D., de Rosnay, P., Takahashi, H., Tang, J., Wang, G., Wang, S., Wang, W., Zhou, X., and Zhu, Y.: Impact of Initialized Land Surface Temperature and Snowpack on Subseasonal to Seasonal Prediction Project, Phase I (LS4P-I): Organization and Experimental design, Geoscientific Model Development Discussions, pp. 1–58, https://doi.org/10.5194/gmd-2020-329, 2021.

[Figure]

Figure 2: Evaluation of simulated snow depth and soil temperatures during 2001–2018. Weighted bias (wBias) of daily snow depth (SND) for ERA5-Land (ERA5L, A) and Exp. MLS-Dis+Den (B). The distribution of SND wBias for ERA5L and each simulation experiment (C). wBias of soil temperature (ST) for 0.07–0.28 m depth for ERAL (D), Exp. MLS-Std (E), and Exp. MLS-Dis+Den (F). Soil temperature for 0.07–0.28 m depth in different permafrost regions (G–K), and winter (DJF) surface offset (SO, K). The number of unique grid cells where observed sites are located is given in the bracket. Color numbers are estimated snow depth and soil temperature bias in winter for observation (OBS) and each simulation experiments.

[Figure]

Figure Snow in TP: Aggregated daily snow depth over the Tibetan Plateau at observed sites (see Figure 2).

[Figure]

Figure 1: Snow compaction rate due to thermal (destructive) metamorphism (excluding liquid water) using different $c_\xi$ in Eq. (6). The dashed line is the percent of ground heat loss through the snow layer in Exp. MLS-Dis+Den compared using the $c_\xi$ of 0.046 (kg m$^{-3}$) in ERA5L, Exp. CTRL, and Exp. MLS-Std (see text for description).

[Figure]

Figure Thermal conductivity: Comparisons of snow thermal conductivity treated in different ways. The influences of water vapour on snow thermal conductivity was estimated by setting snow temperature as -10 °C.

---

## Author Response (AR2)

**Author's Responses to the comments on *"Brief communication: Improving ERA5-Land soil temperature in permafrost regions using an optimized multi-layer snow scheme"**

Bin Cao[1], Gabriele Arduini[2], Ervin Zsoter[2,3]

[1]National Tibetan Plateau Data Center, Institute of Tibetan Plateau Research, Chinese Academy of Sciences, Beijing, China
[2]European Centre for Medium-Range Weather Forecasts, Reading, UK
[3]Department of Geography and Environmental Science, University of Reading, Reading, UK

**Correspondence**: Bin Cao (bin.cao@itpcas.ac.cn)

The authors would like to thank the editor for the thorough assessment of the manuscript. Below we provide a point-to-point response to each comment, editor comments are given in black, responses are given in blue. Additionally, we have included details of how we addressed these changes in the revised submission.

- Line 6: change to "...across the 522 observing stations..."
  Response: revised

- Line 119: change to "...and the period 1979–2000 was used..."
  Response: revised

- Line 124: double check the number of stations provided here compared to the abstract?
  Response: There are 639 stations used in total, among which 522 station are in high-latitudes (as described in the abstract), the others are from high-altitudes.

- Line 126: "The snow performance was evaluated via snow depth at sites where observations are available." Can you specify the number of sites?
  Response: revised as *"...at 173 sites..."*

- Line 149: suggest removing "remarkable"
  Response: removed

- Line 171: change to "...which does not allow..."
  Response: Revised

- Figure 2 caption: change "ERAL" to "ERA5L" in the third line
  Response: Revised

- Thank you for adding the additional information in Figure 2, and the illustrative time series in the supplement. I would like to see a short paragraph added to the supplement which explains the figure. This can be very brief, but would help ensure there is full value to the supplement.
  Response: We added below paragraph to clarify.

  *Text S1*
  *Figure S1 showed the comparisons of observed snow depth and soil temperatures time-series with simulations at selected sites from different geographic regions. The results indicate the snow depth overestimation was reduced in Exp. MLS-Dis+Den with multi-layer snow scheme, and leads to a better representation of soil temperatures with lower bias compared to ERA5-Land.*